# Protein Quality Control in Glioblastoma: A Review of the Current Literature with New Perspectives on Therapeutic Targets

**DOI:** 10.3390/ijms23179734

**Published:** 2022-08-27

**Authors:** Angela Rocchi, Hassen S. Wollebo, Kamel Khalili

**Affiliations:** Center for Neurovirology and Gene Editing, Department of Microbiology, Immunology, and Inflammation, Katz School of Medicine at Temple University, 3500 N. Broad Street, Philadelphia, PA 19140, USA

**Keywords:** protein quality control, glioblastoma, proteostasis, apoptosis, autophagy, treatment-resistant cancer, cancer, ubiquitin proteasome system, autophagic–lysosomal pathway

## Abstract

Protein quality control allows eukaryotes to maintain proteostasis under the stress of constantly changing conditions. In this review, we discuss the current literature on PQC, highlighting flaws that must exist for malignancy to occur. At the nidus of PQC, the expression of BAG1-6 reflects the cell environment; each isoform directs proteins toward different, parallel branches of the quality control cascade. The sum of these branches creates a net shift toward either homeostasis or apoptosis. With an established role in ALP, Bag3 is necessary for cell survival in stress conditions including those of the cancerous niche (i.e., hypoxia, hypermutation). Evidence suggests that excessive Bag3–HSP70 activity not only sustains, but also propagates cancers. Its role is anti-apoptotic—which allows malignant cells to persist—and intercellular—with the production of infectious ‘oncosomes’ enabling cancer expansion and recurrence. While Bag3 has been identified as a key prognostic indicator in several cancer types, its investigation is limited regarding glioblastoma. The cochaperone HSP70 has been strongly linked with GBM, while ALP inhibitors have been shown to improve GBM susceptibility to chemotherapeutics. Given the highly resilient, frequently recurrent nature of GBM, the targeting of Bag3 is a necessary consideration for the successful and definitive treatment of GBM.

## 1. Glioblastoma

Although the presentation and pathogenesis of each glioblastoma varies, there are several key flaws that must exist for abnormalities to become disease. PQC oversight in a healthy system quickly addresses genomic and proteomic errors as they occur. Here, we review the foundational literature on both glioblastoma and PQC. We then discuss the perspective of GBM as a proteinopathy, thus highlighting promising therapeutic targets which aim to restore proteostasis through endogenous molecular oversight.

The most common, most rapidly progressing central nervous system tumor, glioblastoma, is a heterogeneous cancer with both interpatient and intratumor variability [1,2]. This diversity has complicated the development of definitive models of both disease course and treatment [3,4]. While some radiographic and hematologic tests have been validated to assess patient prognosis, no clinical test has replaced diagnosis via invasive biopsy. Here, we provide background information including a description of the typical course of the disease from diagnosis to treatment.

### 1.1. Diagnosis

The incidence of primary glioblastoma increases with advanced age and is higher in the male gender, whereas secondary GBM has been observed in patients with a history of radiation therapy, metastasis, and/or tumorigenic disorders (i.e., neurofibromatosis) [5,6,7]. The most common chief complaints on initial presentation are cognitive impairment and/or seizure [8]. Over the course of the disease, symptoms expand to include motor and sensory dysfunction, drowsiness/fatigue, and headache. Reports of gastrointestinal distress, alopecia, and anorexia increase during systemic treatments (i.e., temozolomide). During end-of-life care, symptoms of dysphagia, dyspnea, and confusion significantly increase in prevalence beyond those of non-GBM palliative-care patients.

GBM diagnosis is made using the WHO classification system—four stepwise criteria of histological and molecular characterization (Figure 1) [9]. Progenitor cell lines of astrocyte or oligodendrocyte origin are identified by exclusionary proteotyping. Only 10–12% of gliomas exhibit mutated IDH resulting in the impaired decarboxylation of isocitrate to αketoglutarate [10,11,12]. This arrest of the Krebs cycle leads to the toxic accumulation of reactive oxygen species, reducing cancer cell resiliency and improving patient prognosis regardless of histology. Mutated samples are therefore excluded from GBM classification.

Patients with wild-type IDH require further histologic and molecular investigation.

GBM classically presents with the highest histologic grade—WHO grade 4—defined by the presence of undifferentiated glia with necrosis and/or microvascular proliferation [9]. Tumors with lower grades 2 and 3—which exhibit poor differentiation but lack necrosis and angiogenesis—are upgraded to GBM if any of three mutations are present: a gain-of-function-mutation in the EGFR promoter, the addition of chromosome 7 in conjunction with the absence of chromosome 10 (+7/−10), and/or the activation of TERT.

EGFR is a transmembrane tyrosine kinase which, when bound by its ligand EGF, initiates the Ras proliferative cascade [13]. EGFR gain-of-function mutations are present in an estimated 45% of GBM [6]. The presence of an extra copy of chromosome 7 has been correlated with the aggressiveness of glioblastomas with increased expression of the mitogens BRAF, EGFR, HOX5A, MET, and PDGFA [13,14,15,16]. Conversely, the loss of chromosome 10 includes the loss of key regulatory genes which promote the tumor-suppressing genes p53, PTEN, and SMAD3/4 and inhibit the tumorigenic gene p52 [16,17,18]. Combined, the +7/−10 pair results in aggressively upregulated cell growth in the absence of tumor suppression. Gain-of-function mutations in TERT enable the ribonucleoprotein polymerase to maintain telomeres, preventing cell senescence and prolonging a cell’s ability to replicate [19]. Any of the two histologic (necrosis, MVP) or three molecular (EGFR, +7/−10, TERT) markers upgrade “IDH wild-type diffuse glioma” to GBM.

### 1.2. Pathogenesis

No single mutation, risk factor, or environmental insult has been causatively linked to GBM onset; however, the pathogenesis of GBM does follow the classic cancer pattern of overexpressed oncogenes and/or under-expressed tumor suppressors (Table 1) [6]. Research in the TCGA identified 12 mutation sites across three regulatory cell cycle cascades. It was found that 73% of all GBM samples displayed mutations in all three pathways, suggesting that the common flaw leading to disease onset is regulatory in nature.

The “pre-metastatic niche” refers to the conversion of a cell’s normal environment into an oncogenic haven [20,21]. Ischemia-induced growth factors promote angiogenesis, while HSF1/Wnt promote fibroblast activation, remodeling of the extracellular matrix, bone marrow production of pro-tumorigenic macrophages, and immunosuppression of anti-tumor lymphocytes [22,23,24].

Although GBM tumors rarely metastasize outside of the CNS, they regularly expand to infect surrounding tissue through oncosomes [25,26]. In addition to transforming the extracellular environment into a pre-metastatic niche, oncosomes enable the infection of intracellular environments of healthy, or less aggressive cancer, cells via macroautophagy. DNA and mRNA have been observed transferring oncogene mutations from cancerous to healthy cells [27]. Proteins, miRNA, and SNPs within exosomes enable the transfer of the proteomic and epigenetic state of tumor cells, advancing the timeline of recipient cell pathology [28,29,30].

### 1.3. Standard of Care

With a five-year survival rate of 5–7% and an average untreated life expectancy of three months, GBM is the third deadliest cancer [31]. This is exacerbated by a non-specific standard of care for what is a significantly individualized pathology [8,31]. First-line therapies raise the average life expectancy by 3 to 11 months, with more than 90% of the patients experiencing recurrence within 6 months of treatment [32,33].

The optimal treatment consists in surgical resection, typically at the time of biopsy, followed by concomitant radiation/chemotherapy and adjuvant therapy with temozolomide ± bevacizumab [33]. Surgical resection of the tumor provides decompressive therapy and slows the progression of GBM invasion by removing the primary insult of the niche. Radiation damages tumor cell DNA, impairing tumor hyperproliferation. Similarly, temozolomide is a DNA-alkylating agent which prevents mitosis by impairing the progression from phase G1 to S [34]. Bevacizumab is a vascular endothelial growth factor antagonist and impairs vascularization, key to tumor sustenance [35].

Modern advancements in epigenetics have the potential to develop the individualized patient care needed to address the uniquities of GBM. Most notably, the epigenetic inactivation of MGMT via methylation of its promoter is necessary for temozolomide sensitivity [36,37]. MGMT is a key DNA mismatch repair protein, the absence of which ensures damage from DNA alkylation will result in cell death.

The molecular investigation of GBM is not standard practice [7,38]. Given the variability of GBM presentation, the ability of multiple neural cell types to exist within one tumor, the consistency of mutation types between recurrences, and the nearly inevitable likelihood of recurrence, the epigenetic interpretation of biopsy samples could improve the current diagnostic model to account for GBM heterogeneity and promote targeted treatments more effectively [39,40].

## 2. Protein Quality Control

PQC is the process by which a cell recognizes and manages abnormal and deformed proteins. In the immediate period, this prevents protein aggregation with healthy structures, reducing the impact of the malformation while restructuring occurs within the complex. In the long term, the complex guides irreparable proteins toward one of two catabolic cascades: the ubiquitin proteasome system and the autophagic–lysosomal pathway. When punctate protein repair is insufficient to offset an insult, full-cell degradation—programmed or otherwise—occurs.

### 2.1. Containment by Heat Shock Protein 70

PQC is initiated by the sentinel chaperone HSP70 [41]. Preserved across eukaryotic evolution, the HSP70 superfamily exhibits three functional domains that are key to PQC: an ATPase, a hydrophilic substrate binding site, and the C-terminal sequence Glutamate–Glutamate–Valine–Aspartate (EEVD) [42]. PQC HSP70s can be simplified into two key factions: maintenance heat shock proteins (m-HSP70) and restoration (r-HSP70) heat shock proteins. The six m-HSPs are constitutively active and regulate nascent proteins through both post-translational modification and guided transport. Impairment in any m-HSP70 is incompatible with cell survival [43].

Transcription of the four r-HSP70s is upregulated in the presence of stress, making them key for cell adaptability. Cell stress physically initiates the common cascade of PQC by inducing a conformational change in Heat Shock Factor 1 directly or via stress, or specific, cell mediators [44]. Once activated, HSF1 enters the nucleus to bind the HSP promoter. The increased r-HSP70 population therefore enables mass refolding and/or the removal of denatured proteins due to the triggering stressor. Although negligible at baseline, knock-out of an r-HSP becomes evident in non-ideal conditions with significantly higher susceptibility to proteinopathy and cell death [45,46].

At baseline, ADP-bound HSP70 exists in a closed state wherein the SBD pocket is self-sealed by its α-helical cap (Figure 2) [47]. The binding of ATP induces an allosteric change, moving the cap about a hinge to obstruct the ATPase, while opening the SBD pocket for substrate binding [42,48]. Once a substrate is bound, another conformational change occurs: the alpha-helix cap is returned to the pocket, hydrolysis is disinhibited, and the energy generated by ATP hydrolysis seals the substrate within the SBD. Another ATP molecule can be utilized to refold the contained substrate; however, it must compete with proteins containing a BAG domain [49].

### 2.2. Direction by BCL2-Associated Anthogenes

The six members of the BCL2-associated anthogene (Bag) family bind the HSP70 ATPase through their eponymous BAG domain [49]. The C-terminal domain was first discovered complexing with anti-apoptotic BCL2. While BAG–BCL2 complexes inhibit the pro-apoptotic BAX/BAK, MOMP, and TNFR, BAG-HSP70 complexes exhibit varied roles, directing poor-quality proteins to different branches of the PQC pathway through their specialized non-BAG domains (Figure 3).

Bag1 exhibits a lysine-rich UBL [50]. With BAG connected to HSP70 and UBL to the 26S proteasome, the substrate protein is delivered to the UPS [51]. The full-length isoform of Bag1 also exhibits a nucleus localization sequence and a DNA binding motif, suggesting a role in the basal PQC of transcription.

Bag2 contains an N-terminal coil that enhances the activity of HSP70 through its overlap with the substrate pocket of HSP70; this accelerates ATP/ADP exchange and facilitates refolding [52,53].

Bag3 is characterized by WW and PxxP domains. The WW domain is essential for the induction of chaperone-mediated autophagy, as it enables the interaction with the lysosome [54]. The PxxP chain of proline repeats interacts with the motor protein dynein [55,56,57]. With BAG connected to HSP70 and PxxP connected to dynein, the contained molecule is transported to LAMP2A where a WW interaction guides ALP. Of note, Bag3 levels are negligible in healthy cells [58]. Its transcription increases exponentially in the presence of the promoter HSF1, with positive feedback by Bag3 increasing the role of autophagy during prolonged stress [59].

Bag4 is also known as SODD (silencer of death domain) due to its affinity for and obstruction of pre-programmed cell death sequences: Tumor Necrosis Factor Receptor 1 (TNF-R1), Death Receptor 3 (DR3) and Fas [60,61]. Prevalent in most tissues, Bag4 prevents the spontaneous activation of cell death cascades by the TNF superfamily. States of inflammation exact the opposite, with decreased Bag4 disinhibiting apoptosis.

Bag5 consists of five BAG domains that appear to inhibit HSP70 and promote cell death [62]. Stress-induced p53 directly promotes Bag5 expression, translocating Bag5 from the cytosol to the ER, where it can assist PQC [63,64]. This mechanism contradicts the pro-death behavior of Bag5 observed in several pathological states.

Bag6 includes several shared domains including PxxP, NLS, and UBL [65]. It also exhibits a unique LIR1 motif that links Bag6 to the ER for insertion of nascent proteins into membranes [66]. Stress induction cleaves Bag6 so that the LIR1 motif obstructs LC3B [67]. LIR1 inhibition of ALP enhances UBL promotion of UPS, shifting the route of PQC.

### 2.3. Degradation by the Ubiquitin Proteasome System

The UPS branch of the PQC cascade is responsible for 80–90% of all pro-survival, punctate protein degradation [68]. Proteins are identified for UPS by a three-step process of ubiquitination: activation by E1 enzymes, conjugation by E2 enzymes, and ligation by E3 enzymes [69]. With the increasing number of family members with each sequential enzyme group, this branching recognition system gradually increases the specificity of protein tagging. It is the final and most specialized enzyme, E3, that determines the substrate that E1/E2-prepared ubiquitin binds. Through its UBL, Bag1 associates with E3, facilitating the ubiquitination of HSP70-bound clients for proteolysis [70,71].

Proteasome 26 is the most populous complex, involving a 19S “cap”—responsible for sieving substrates—and a 20S “core”—responsible for active proteolysis. The 19S regulatory protein non-ATPases, or Rpns (most notably, Rpn1, Rpn10, and Rpn13), recognize Ub-tags with specific chain lengths and branches [72,73]. This bond is strong but temporary, reversible, and ATP-independent; it creates a noncommittal “dwell time” for protein quality inspection. In these milliseconds, quick deubiquitinases (Usp14 and Uch37) can prevent irreversible proteolysis by removing the Ub-tags before permanent changes can be made by the proteasome core [74,75].

There are many checkpoints in the UPS prior to the expenditure of energy and the commitment to unfolding/cleavage. Ubiquitination is balanced by deubiquitinases, the cap-Ub-tag bond is purely ionic, and the pore structure requires a loose, poor-quality protein to unfold. Only when these conditions are met, does irreversible proteolysis occur [76].

Once the 19S cap has held a client longer than the dwell time, ATPase activity of 19S induces a physical change that exposes the peptide chain to slow the ubiquitinases and widens the catalytic pore. Rpn11—the slow ubiquitinase that quick ubiquitinases must out-compete in order to prevent degradation—then separates the final Ub-tag from the N-terminus, enabling the threading of the substrate through the molecular shredder [77]. This hydrolysis enables an irreversible bond formation between the poor-quality protein and the pore of the protease complex, so hydrolysis can occur.

### 2.4. Degradation by the Autophagic–Lysosomal Pathway

The ALP is less active than the UPS, accounting for only 10–20% of punctate protein removal in PQC [68]. There are three parallel ALP types: chaperone-mediated autophagy, macroautophagy, microautophagy. Of these mechanisms, only CMA is selective and condition-sensitive [78,79]. Favored in high-proliferation, high-stress environments, selective ALP is essential for the pluripotency of embryonic stem cells and the function of adult stem cells [80,81]. Similarly, during pathogenesis, CMA is essential to cell survival—accounting for 30% of cytosolic PQC, with its absence leading to proteotoxicity and apoptosis [82,83,84].

The WW domain of Bag3 chaperones HSP70 to p62 on the lysosome membrane where HSP70 binds the external tail of lysosome-associated membrane protein [74,78,79,82]. LAMP2A is the rate-limiting step of CMA, as HSP70 threads the unfolded protein through its pore, into the lumen. The AKT/mTOR system regulates the phosphorylation/deactivation status of LAMP2A, increasing the stress status selectivity of Bag3–HSP70-mediated autophagy [85]. Unique to chaperone-mediated autophagy, the conditions of the docking receptor LAMP2A, the protein sequence K–F–E–R–Q (or KFERQ-like domain), and a loosely folded protein must be present for HSP70 to release the client to the lysosome. Once transferred through the membrane, a protein is degraded by lysosomal hydrolases.

## 3. Glioblastoma as a Proteinopathy

Cancer cells disobey the standards of PQC. When an aberrant protein is identified by HSP70, HSP70 must release dormant an RNA-like endoplasmic reticulum kinase to sequester the deformity [86]. No longer inactivated by HSP70, PERK proceeds to generally suppress eIF2α and selectively promote ATF4. The resultant inhibition of protein synthesis by eIF2α conserves energy and halts cell growth; acute ATF4 activation both promotes autophagy and inhibits apoptosis [87]. During prolonged periods of stress, this pathway fatigues: disinhibited eIF2α enables protein synthesis in an already proteotoxic environment, and ATF4 transcription shifts to promote, rather than prevent, apoptosis [88,89]. In GBM, cells manage to proliferate in proteotoxic environments by utilizing the PERK/eIF2α/ATF4 pathway [90]. The exact mechanism by which this occurs has not been established; however, inhibitors of PERK have been shown to sensitize cancer cells to chemotherapeutics. Here, we discuss the ways in which cancers have been linked to PQC processes to better support the hypothesis that glioblastoma may be treatable at the level of the Bag directional proteins.

### 3.1. Cancer and the Common Cascade

Due to its role in protein fating, HSP70 has a duplicitous reputation in oncology. Well reviewed by Vostakolaei and peers, HSP70 is critical to cancer prevention and survival [91]. In early cancers, HSP70 is upregulated. Whether this is a defect in HSP70 itself or a response to cancer stressing its environment can be case-dependent, but its behavior is consistent. If proteostasis is restored, PQC and HSP70 are successful. If the insult is too great, AIF outcompetes HSP to trigger apoptosis. If oncogenesis occurs, HSP70 remains loyal to the cell—maintaining a new standard of proteostasis that supports the now-malignant cell cycle. This makes HSP70 a strong biomarker for the diagnosis, prognosis, and treatment of several tissue types.

HSP70 has been implicated in the initiation, migration, and invasion of tumors of the breast, cervix, gastrointestinal tract, lungs, skin, lymphocytes, and brain [92]. Conversely, HSP70 inhibition has shown therapeutic benefit as both a chemotherapy co-treatment and an independent therapy in many clinical trials [93]. Despite these advances, no HSP70-specific inhibitor has made it to market. Telaprevir is an approved equivalent in the setting of infectious disease. By obstructing the ATPase of DnaK (the bacterial analog of HSP70), telaprevir interrupts proteostasis and has been shown to reverse bacterial resistance to the anti-mycobacterial rifampin [94]. No such medication has entered oncologic practice.

### 3.2. Cancer and the Directional Proteins

The BAG family acts athánatos—the Greek term “against death” for which these proteins are named [95]. Cancer cells inherently share this behavior, and multiple cancer types have been linked to the Bag proteins (Table 2) [96,97,98,99,100,101,102,103,104,105,106,107,108,109,110,111,112,113,114,115,116,117,118,119,120,121,122,123]. Given the myriad ways in which Bag1–6 and their cochaperones have been observed in oncology, the therapeutic potential of PQC manipulation at the level of the directional proteins is not limited to glioblastoma. Instead, the data collected in Table 1 and Table 2 indicate disturbances in protein quality control are necessary for cancers to survive. Similarly, the corrections of these disturbances can impair current cancers and prevent future cancers regardless of the degree of heterogeneity.

While many pharmaceuticals target the downstream pathways of UPS and ALP, the anthogenes are predominantly explored as prognostic indicators rather than as interventions [99,101,102,104]. Further investigation is warranted to enhance pre-existing therapies at the level of anti-apoptotic Bag3, which promotes cell survival early in the PQC decision-making process.

### 3.3. Cancer and the Ubiquitin Proteasome System

Proteasome inhibitors are presently utilized in the treatment of hematologic neo-plasms such as multiple myeloma and B-cell lymphoma [124]. Boronates, bortezumib, and ixazomib reversibly obstruct the 20S core, while epoxyketones, carfilzomib, and oprozomib irreversibly bind and disable the active site.

Unfortunately, addressing only one limb of the PQC cascade creates an opportunity for resistance, and the decline in proteolysis can be compensated by an increase in autophagy—observed at the directional chaperone level as a significant increase in Bag3 expression [125]. The compensatory response of ALP in the absence of the UPS has been circumvented with the knock-out of the Bag3 gene or the inhibition of the autophagy stimulator sphingosine kinase 2 [126,127], suggesting that repression of autophagy is ultimately necessary to ensure consistent cell death.

### 3.4. Cancer and the Autophagy–Lysosome Pathway

Autophagy has been correlated with general longevity [128]. This is true for both healthy and cancer cells, with high autophagy markers observed in higher-grade gliomas with poorer patient outcomes [129,130,131,132]. ALP enactors increase markedly with age and GBM progression. ALP inhibition improves the anti-tumor immune response, decreasing metastatic adhesion and increasing tumor cell apoptosis [133].

In proteotoxic conditions, such as cancer, the transcription factor EB redirects the ALP pathway from lysosome degradation to exocytosis [134,135]. In the nucleus, TFEB increases the production of lysosome and autophagosome membranes [136]. Along the plasma membrane, TFEB activates MCOLN1, a calcium channel responsible for synaptotagmin VII-associated exocytosis [137]. The resultant clearance of the cell contents can compensate for an overwhelmed organelle-based degradation; however, it enables the spread of the pathology [138,139].

Indicators of resilience and adaptability, autophagy markers are also upregulated in glioblastoma cells following the treatment with chemotherapeutics and during recurrence [140]. For this reason, autophagy markers have been validated in multiple prognostic risk scoring systems, while autophagy inhibitors have become promising cancer treatment options [140,141,142].

In multiple neuroblastoma models, the Corallo group successfully reversed drug resistance to the tyrosine kinase inhibitor ponatinib by co-treating with the autophagy inhibitor chloroquine [143]. Repeated glioblastoma clinical trials agree with this finding, with the addition of chloroquine to a pharmacological regimen correlating with an increased life expectancy for patients [144,145]. The combination chloroquine/hydroxychloroquine, which was first FDA-approved for its anti-malarial properties, inhibits the fusion of the autophagosome and the lysosome [146].

The beclin-1 inhibitor spautin-1 has been shown to sensitize chronic myeloid leukemia cells to another tyrosine kinase inhibitor, imatinib [147]. Further investigation of spautin-1 properties is warranted in the setting of glioblastoma and first-line treatment temozolomide, a DNA alkylating agent.

Regorafenib, a medication approved for colorectal cancer treatment, has been observed to indirectly obstruct autophagolysosome fusion. Successfully halting macroautophagy, teams have found that regorafenib induces apoptosis in recurrent and treatment-resistant glioblastoma multiforme [85,148].

Bafilomycin, approved for use as a macrolide antibiotic, inhibits autophagy downstream, by impairing the influx of protons through vacuolar H+ ATPase, removing the acidity-dependent degradation of luminal contents, and forming toxic protein aggregates [149,150].

JG-98, which is currently in clinical trials, impairs Bag3 from complexing with HSP70 [151]. Of note, the Yaglom team recognized JG-98 as a collaborative therapy, identifying inhibitors of the proteasome that synergize with its anti-autophagic behavior, therefore anticipating the compensatory reflex of PQC. Unfortunately, this treatment has shown signs of toxicity in non-brain, high-stress tissues that heavily rely on PQC, including cardiomyocytes and skeletal muscles [152].

## 4. Conclusions

For homeostasis to exist, allowing a cell to remain healthy, the natural PQC pathway must be optimized. Alteration of its regulators—namely, Bag3 and its cochaperones—enables disease in both excess and deficiency. Research must continue into the refinement of small molecules’ properties in the PQC cascade to better model heterogeneous and treatment-resistant diseases, like glioblastoma.

Targeting Bag3 at the advent of PQC postulates a unified model for GBM. Complicated by high intra- and interpatient variability, the treatment resistance and recurrence of GBM limits therapies and worsens prognoses. [153]. Despite these challenges, there are two PQC mechanisms that can be corrected in a majority of GBM types: intracellular anti-apoptosis and extracellular immunosuppression. While we have discussed myriad potential targets in the PQC pathway, Bag3 is the most promising candidate, as it connects these two processes.

Self-promoting, anti-apoptotic, and stress-induced Bag3 redirects the HSP70–substrate complex away from the proteosome and toward the lysosome [54,55,56,57,58]. This increases the relevance of Bag3 compared to other Bag proteins, as inhibition of apoptosis by autophagy is central to tumor resilience [154]. Bag3 directly promotes pro-survival PERK and pro-proliferation EGFR [58,155,156]. Bag3 also inhibits tumor suppressors p53 and Rb, enabling cell survival and growth despite ER stress [157,158]. Remarkably, this implicates Bag3 in each of the three metabolic pathways most commonly mutated in GBM (Table 1) [6].

Extracellular protein quality control is regulated by the immune system; however, the pre-cancerous niche is both immunosuppressive and tumor-promoting [21,22,23,24,25,26,142]. Even if intracellular PQC is successfully restored, the potential of recurrence is statistically inevitable due to the infectious conversion of apoptotic blebs into oncosomes [27,159]. Exosomes—in PQC clearance or apoptosis—are linked to Bag3 in two ways. Bag3 has been identified as a loading molecule of exosomes, while p53, which Bag3 suppresses, is key in the production of immunostimulatory exosome membranes [160,161,162]. The inhibition of exosome production inhibits GBM proliferation, and GBM proliferation is associated with increased secretory autophagy [163]. Reducing tumorigenic communication via oncosome suppression therefore has potential in both treatment of present cancer and prophylactic inhibition of recurrence, a notable contributor to the short lifespan of patients with GBM [32].

The redundancy of two PQC branches and myriad regulatory molecules highlights the importance of PQC to our survival. It also translates to a predominant challenge in cancer treatment. Presenting as metastasis and resistance to treatment, PQC dysregulation must be addressed for cancer therapies to maintain viability [3]. Given the prevalence of drug resistance, the observed ALP/UPS compensation, and the predominance of recurrence, the intracellular and extracellular regulator Bag3 offers a stress-sensitive and, therefore, cancer-cell specific target to effectively re-sensitize malignant cells to PQC oversight, even in the presence of a heterogeneous molecular profile.

The development of small molecules that can selectively activate or suppress Bag3 expression and, at the same time, modulate Bag3 interaction with partner proteins can serve as a possible therapeutic avenue. The controlled expression of Bag3 is essential for its biological function; therefore, it is imperative to understand how Bag3 gene expression is regulated in cells, tissues, and disease-associated pathological conditions.

## Figures and Tables

**Figure 1 ijms-23-09734-f001:**
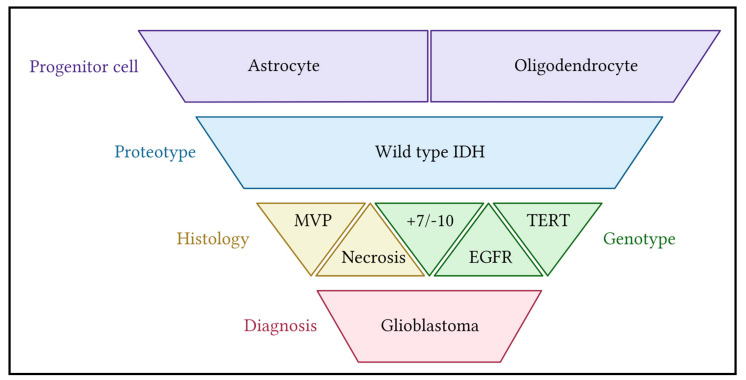
Shown are the 2021 WHO classification criteria for glioblastoma. In order of increasing specificity (top to bottom): (1) Biopsy assessment begins with the identification of the glial progenitor cell line. (2) Cells with mutant IDH are excluded. (3) Tumors with necrosis or MVP on histology are grade 4 and diagnosed as glioblastoma. Three mutations are also included in the GBM diagnosis, regardless of histology: the addition of a whole chromosome 7 in the absence of chromosome 10, TERT activation or EFGR gain of function [9].

**Figure 2 ijms-23-09734-f002:**
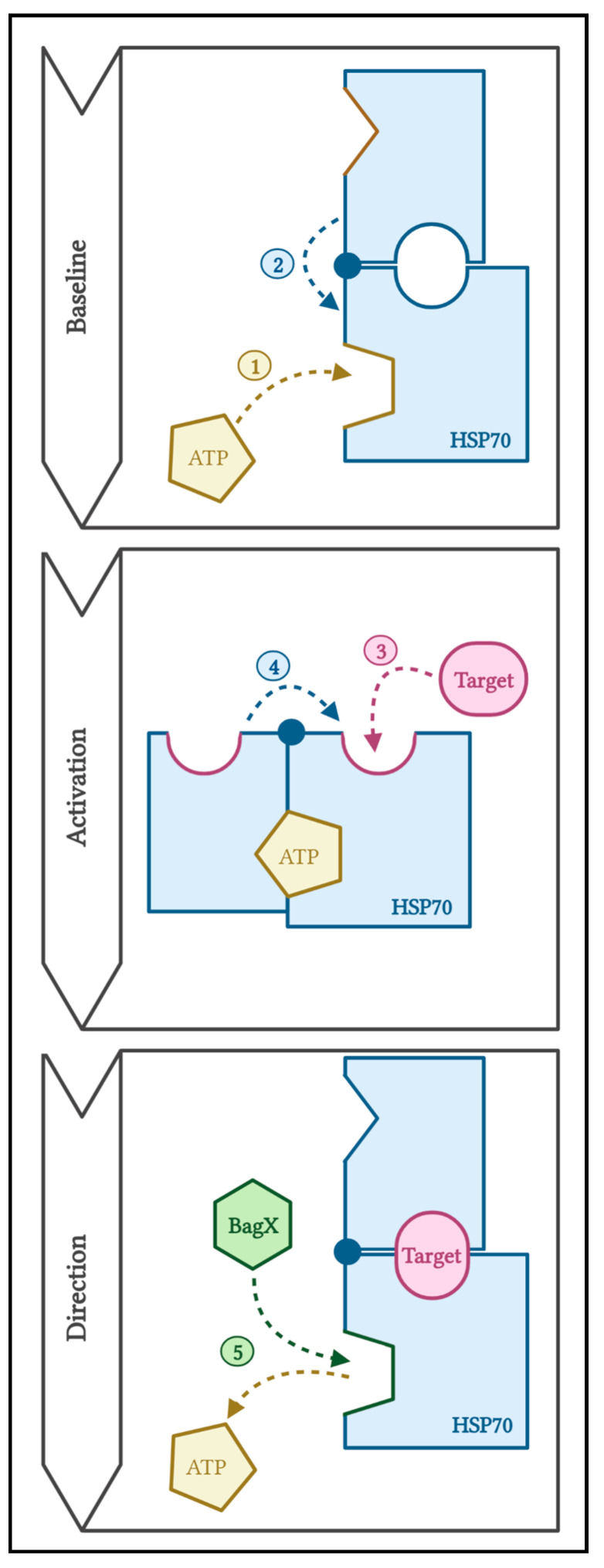
Mechanism of function of HSP70. (1) ATP binds ATPase. (2) A conformational change in the hinged cap inhibits the ATPase and exposes the SBD. (3) An aberrant protein binds SBD. (4) A conformational change traps the substrate. (5) Directional proteins compete for the ATPase binding site.

**Figure 3 ijms-23-09734-f003:**
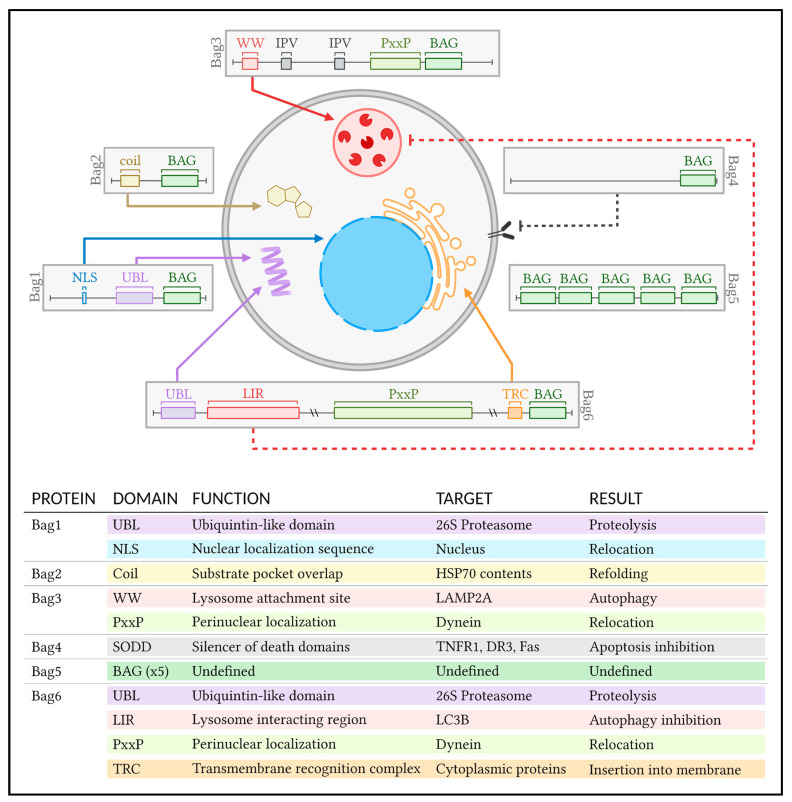
Roles of BCL2-associated anthogenes and their associated functional domains.

**Table 1 ijms-23-09734-t001:** Prevalence of oncogenic mutations and expression abnormalities in glioblastoma biopsy samples in the Cancer Genome Atlas.

Mutation	Prevalence	Impact
Ras proliferation	88%	Mitosis
EGFR *	45%	Amplification of growth receptor
PTEN *	36%	Deletion of P13K inhibitor
NF1	23%	Silencing/deletion of Ras suppressor
PI3K	15%	PI3K gain of function
PDGFRA *	13%	Amplification of growth receptor
p53 tumor suppression	87%	Persistence of oncogenes
ARF	49%	Deletion of p53 dis-inhibitor
p53	35%	p53 silencing/deletion
MDM2	14%	Amplification of p53 suppressor
Rb tumor suppression	78%	Disinhibition of G1/S progression
CDK N2A	52%	Deletion of Rb dis-inhibitor
CDK N2B	47%	Deletion of Rb dis-inhibitor
CDK4	18%	Amplification of Rb inhibitor
Rb	11%	Deletion of Rb

* Associated with an increase in grade of glioblastoma secondary to +7/−10 chromosomal abnormality.

**Table 2 ijms-23-09734-t002:** Abnormal BAG gene and Bag protein levels that have been associated with cancer(s).

Study	Biopsy Source	Outcome
Bag1		
Aveic et al., 2011 [96]	Pediatric bone marrow	Bag1 protein is required for acute myeloid lymphoma cell survival.
Bai et al., 2007 [97]	Colon carcinoma	Bag1 gene expression is strongly associated with metastasis, shorter survival, and advanced staging of colon cancer.
D’Arcangelo et al., 2018 [98]	Melanoma and nevi ^(HPA)^	Bag1 gene and protein expression are diagnostic for melanoma, with levels differentiating malignant and benign nevi.
Du et al., 2021 [99]	Breast cancer ^(TCGA)^	Bag1 gene expression is prognostic for severity and outcomes of breast cancer.
Gennaro et al., 2019 [100]	Osteosarcoma *	Isoform Bag1S inhibits MYC-induced apoptosis, promoting cancer cell survival.
Mariotto et al., 2021 [101]	B-cell acute lymphoblastic lymphoma (Zebrafish)	Bag1 is prognostic for severity and outcomes of B-ALL. Bag1 inhibitor Thio-2 induces cytotoxicity as a sole therapy and increases the pro-apoptotic effects of other B-ALL therapies.
Wu et al., 2021 [102]	Renal clear cell carcinoma ^(TCGA)^	Bag1 is prognostic for severity and outcomes of RCCC.
Bag2		
Hong et al., 2018 [103]	Esophageal squamous cell carcinoma ^(TCGA)^	Bag2 overexpression is predictive of poor survival outcomes in ESCC.
	Esophageal squamous cell carcinoma *	Bag2 knock-out inhibits ESCC proliferation.
Sun et al., 2020 [104]	Gastric cancer ^(HPA)^	Bag2 protein levels correlate with gastric cancer prognosis.
	Gastric cancer *	Bag2 enhances proliferation and metastasis of tumor cells.
Yue et al., 2015 [105]	Bone, liver, colorectal, breast, and lung cancers *	Bag2 maintains mutant p53, increasing its gain-of-function cell growth, metastasis, and treatment resistance.
Zhang et al., 2021 [106]	Hepatocellular carcinoma	Bag2 is significantly upregulated in HCC, with higher levels correlating with shorter survival.
	Hepatocellular carcinoma *	Silencing of Bag2 facilitated apoptotic intervention, improving HCC treatments.
Bag3		
Shi et al., 2018 [107]	Chondrosarcoma	Bag3 expression is significantly increased in malignant chondrosarcoma compared to normal cartilage and benign tumors.
Lee et al., 2019 [108]	Gastric cancer	Bag3 is upregulated in response to hepatocyte growth factor, increasing cancer resistance, proliferation, and invasion.
Li et al., 2018 [109]	Colorectal cancer	Bag3 levels correlate with patient gender and tumor size.
	Colorectal cancer *	Bag3 gene knock-out impairs proliferation, metastasis, and chemoresistance of colorectal cancer cells.
Linder et al., 2022 [110]	Glioblastoma *	Bag3 inhibits ciliogenesis, increasing GBM aggression and treatment resistance.
Wang & Tian, 2018 [111]	Cervical cancer	Inhibition of Bag3 halts cell proliferation and metastatic invasion of cervical cancer.
Yunoki et al., 2015 [112]	Retinoblastoma *	Bag3 protects retinoblastoma cells from apoptosis in the setting of heat stress.
Bag4		
Du et al., 2015 [113]	Hepatocellular carcinoma *	Bag4 expression increases proliferation and survival of liver cancer cells.
Jhang et al., 2021 [114]	Gastric cancer	Bag4 expression levels correlate with stage, metastasis, tumor size, and outcomes in patients with gastric cancer.
Rho et al., 2018 [115]	Blood plasma pre- and post-diagnosis of colon cancer	Bag4 elevation can be used in an early-detection biomarker panel for the screening/diagnosis of colon cancer.
Bag5		
Bi et al., 2016 [116]	Ovarian cancer *	Bag5 is tumorigenic in epithelial ovarian cancer with Bag5 knock-down improving the effect of tumor-suppression therapy.
Bruchmann et al., 2013 [117]	Prostate cancer	Bag5 is overexpressed in prostate cancer with stress-induced migration to the ER inhibiting apoptosis.
Che et al., 2021 [118]	Hepatocellular carcinoma	Endogenous Bag5 inhibitor PRMT 6 (protein arginine N-methyltransferase 6) is decreased in HCC, disinhibiting Bag5, stabilizing HSP70, & increasing pro-survival ALP.
	Hepatocellular carcinoma *	Suppression of Bag5 leads to instability of HSP70, increasing cell susceptibility to anti-cancer agents (i.e., sorafenib).
Wang et al., 2021 [119]	Ovarian tissue following cisplatin trial	Bag5 knockdown contributes to the metabolism shift of cancer development. Bag5 is increased treatment-sensitive ovarian cancer.
Yan et al., 2021 [120]	Hepatocellular carcinoma *	Bag5 expression promotes hepatocellular oncogenesis.
Zhang et al., 2020 [121]	Papillary thyroid cancer *	Bag5 is overexpressed in papillary thyroid cancer & is requisite for invasion & metastasis.
Bag6		
Ragimbeau et al., 2021 [122]	Colon cancer *	Bag6 protein is requisite for colon cancer cells to proliferate.
Schuldner et al., 2019 [123]	Melanoma *	The Bag6-p53 pathway is requisite for pro-oncogenic exosome formation.

* Primary cell lines; ^(HPA)^ Human Protein Atlas; ^(TCGA)^ The Cancer Genome Atlas.

## Data Availability

Not applicable.

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
