# Peer review of "Protein Quality Control in Glioblastoma: A Review of the Current Literature with New Perspectives on Therapeutic Targets"

_ijms, 2022, doi:10.3390/ijms23179734_

Round 1

Reviewer 1 Report

The authors explore the role of proteins BAG1-6 in protein quality control mediated by HSP70 and cancer with special focus in glioblastoma.

1. The manuscript has no mention of ubiquitin linkage K48 and K63, their difference and role in targeting proteins to different type of degradation.

2. The title is misleading. Rather than being a review on cancer, this review focuses on glioblastoma. Similarly, rather than being a review on protein quality control, this review focuses on BAG1-6. Consequenctly the title should contain the words glioblastoma and BAG1-6.

3. line 279, the concept of directional proteins is entirely underdeveloped in this review. This paragraph should be expanded to explain the concept.

4. line 32, clinical is misspelled

5. line 4. Titltes of authors appear as authors.

Author Response

Reviewer #2:

  1. The manuscript has no mention of ubiquitin linkage K48 and K63, their difference and role in targeting proteins to different type of degradation.

The specific roles of K48 and K63 were included among other ubiquitination steps in section 2.3. Given the extensive complexity of the myriad sub-sections of ubiquitination, we limited our review to the foundational concepts as they pertain to the stress-induced Bag/HSP oversight system.

  1. The title is misleading…

To better reflect the contents of our review we have changed the title to “Protein Quality Control in Glioblastoma: a review of the current literature with new perspectives on therapeutic targets”

  1. Line 279: the concept of directional proteins is entirely underdeveloped in this review. This paragraph should be expanded to explain the concept.

We disagree that the directional proteins are underdeveloped, however, the in-depth analysis of each Bag protein was moved from Figure 3 into the main text while a summary table replaced it in the subtitle.

  1. Line 32: clinical is misspelled.

We were unable to identify the exact error reported, however, we have re-reviewed the text in its entirety for typographical and grammatical errors.

Reviewer 2 Report

The topic of the review is interesting especially because to date, we have not yet been able to understand which potential targets in glioblastoma could be exploited in therapy in the treatment of such an aggressive and heterogeneous tumor.  

My perplexity and confusion lie in the fact that the authors write an introduction on the general characteristics of glioblastoma but then address the issue of PQC in a completely general way, referring to different tumor models and not focusing on glioblastoma. The title itself is too general.

In this way several recent papers describing molecular chaperones, ubiquitin/proteasome-dependent protein degradation and autophagy machinery in glioblastoma are not discussed (for example: doi: 10.1002/cjp2.134; doi.org/10.3389/fonc.2020.574011; doi.org/10.1186/s41016-020-00211-3; DOI: 10.1126/scisignal.aal2323; doi: 10.3390/ph13070156; DOI:10.1007/s12035-021-02339-4).

This should be the main purpose of the review.

The authors should therefore first clarify what purpose they have: PQC in cancer or in glioblastoma? And then reorganize and complete the review accordingly

Minor points:

-Line 123 authors should better describe the multiple cell types to exist within one tumor (mesenchymal, neural …)

-Check for Typos (line 32 “cliical”)

-Line 96: Authors should replace the word infect with invade in the sentence “to infect surrounding tissue…”

-Line 110; What is the meaning of the symbol used?

Author Response

   My perplexity and confusion lie in…referring to differnet tumor models and not focusing on glioglastoma.

The goal of our review is to highlight the potential involvement of protein quality control (PQC) in a broad range of cancer that may also apply to glioblastoma as a proteinopathy. We offere several proposed models of other cancers to educate hypothesis and stimulate discussion regarding PQC as a model of the currently unmodeled brain cancer. We  hope that our discussion in this review will inspire glioblastoma-specific investigations as the perspective discussed is yet to be fully explored.

Ÿ…several recent papers describing molecular chaperones, ubiquitin/proteasome-dependent protein degradation, and autophagy machinery in glioblastoma are not discussed for example…

We appreciate the many publications recommended; however, we do not want to overextend the breadth of the review. For example:

*        doi: 10.1002/cjp2.134

The 2019 paper by Mellai et al. investigates SEL1L as a key player in protein degradation of the endoplasmic reticulum. The role of the endoplasmic reticulum PQC pathway differs from that discussed in this article. Although similar, the ER is key in the initial formation of proteins whereas the UPS and ALP systems discussed are predominantly responsive to insult.

*        doi: 10.3389/fonc.2020.574011

The 2020 review by Scholz et al. discusses the value of ubiquitin-targeted therapies for glioblastoma. This is discussed in section 3.3. and includes shared citations of original research publications.

Rather than incorporating these and other papers in GBM and PQC literature, we have taken the reviewer’s advice to narrow our title to better reflect the limitations of our paper.

Ÿ          The authors should therefore first clarify what purpose they have…

To better clarify the layout of our review, we incorporated a short introductory statement addressing the foundational background of GBM and PQC prior to the combined discussion of therapeutic relevance. We have also extended the introduction of each major section to maintain the conceptual thread.

Minor points:

  • Line 123: authors should better describe the multiple cell types to exist within one tumor (mesenchymal, neural…)

We found this sentence on line 147 and added the recommended “neural” adjective to clarify the common progenitor status.

  • Line 96: authors should replace the word infect with invade in the sentence “to infect surrounding tissue”

We found the sentence of concern on line 124. Although the connotation of the word “infect” is typically limited to microbial pathogenesis, we consider it suitable for this context. While a proliferating tumor mass can “invade” surrounding tissue in displacing neighboring structures; this is not a result of the process described.  Oncosomes transfer poor quality proteins from cancer cells to healthy ones, resulting in the conversion of glia into glioblastoma. Thus, we believe GBM spread via oncosomes is best described with the word “infect.”

  • Line 110: What is the meaning of the symbol used?

We were unable to identify the symbol referenced. Potential confusion regarding the asterisk in Table 1 was addressed through the reformatting of the table such that the footnote is more evident. We also found several symbols for alpha (α) miscoded, resulting in its replacement with an error indicator (a). These were corrected in lines 151, 152, 153, and 156.

  • Check for typos (Line 32 “cliical”)

We were unable to identify the exact error reported, however, we have re-reviewed the text in its entirety for typographical and grammatical errors.

Round 2

Reviewer 2 Report

Authors replied exhaustively to my requests.